# Validation of Sputum Biomarker Immunoassays and Cytokine Expression Profiles in COPD

**DOI:** 10.3390/biomedicines10081949

**Published:** 2022-08-11

**Authors:** Alex Mulvanny, Caroline Pattwell, Augusta Beech, Thomas Southworth, Dave Singh

**Affiliations:** 1Division of Immunology, Immunity to Infection and Respiratory Medicine, School of Biological Sciences, Faculty of Biology, Medicine and Health, Manchester Academic Health Science Centre, The University of Manchester and Manchester University NHS Foundation Trust, Manchester M13 9PL, UK; 2Medicines Evaluation Unit, Manchester University NHS Foundation Trust, Manchester M23 9QZ, UK

**Keywords:** chronic obstructive pulmonary disease, validation, biomarker, sputum, inflammatory endotype, airway colonisation

## Abstract

Immunoassays are commonly used to assess airway inflammation in sputum samples from chronic obstructive pulmonary disease (COPD) patients. However, assay performance and validation in this complex matrix is inconsistently reported. The aim of this study was to assess the suitability of various immunoassays for use with sputum samples, followed by use of validated immunoassays to evaluate biomarker levels in COPD patients. Assays were assessed for recombinant reference standard suitability, optimal sample dilution, standard recovery in the biological matrix and reproducibility. Validated assays were used to assess sputum supernatants in Cohort A (*n* = 30 COPD, *n* = 10 smokers, *n* = 10 healthy) and Cohort B (*n* = 81 COPD, *n* = 15 smokers, *n* = 26 healthy). Paired baseline and exacerbation samples from 14 COPD patients were assessed in cohort A, and associations with sputum cell counts and bacterial colonisation investigated in cohort B. 25/32 assays passed validation; the primary reason for validation failure was recombinant reference standard suitability and sample dilution effects. Interleukin (IL-)6 and IL-8 were significantly increased in COPD patients compared to healthy subjects and smokers for both cohorts. Tumour necrosis factor (TNF)α and IL-1β were higher in COPD compared to smokers using one immunoassay but not another, partly explained by different absolute recovery rates. IL-1β, IL-2, IL-4, IL-8, IL-17A, Granulocyte colony stimulating factor (G-CSF), Interferon (IFN-)γ, Interferon gamma induced protein (IP-)10, Macrophage inflammatory protein (MIP)-1α, MIP-1β and TNF-α levels correlated with sputum neutrophil percentage in COPD patients. IL-1β, IL-4, IL-8, G-CSF and IFN-γ levels were associated with *Haemophilus influenzae* colonisation in COPD patients. Current smokers had lower levels of IL-1β, IL-4, IL-8, G-CSF, IFN-γ, IP-10, Monocyte chemoattractant protein (MCP)-1, MIP-1α, MIP-1β and TNF-α. Validated immunoassays applied to sputum supernatants demonstrated differences between COPD patients and controls, the effects of current smoking and associations between *Haemophilus influenzae* colonisation and higher levels of selected cytokines. Immunoassay validation enabled inflammatory mediators associated with different COPD characteristics to be determined.

## 1. Introduction

Chronic obstructive pulmonary disease (COPD) is characterised by persistent airway inflammation and airflow limitation [1]. COPD is a heterogeneous condition, with considerable variation observed between individuals in both the clinical manifestations and the pathophysiological features present [2,3]. Consequently, it is recognised that different management strategies may be required for COPD subgroups based on clinical characteristics (clinical phenotype) and/or biological characteristics (endotype) [2,3,4,5]. Airway inflammation in COPD patients involves a complex network of inflammatory cells, cytokines and chemokines. The measurement of the nature and burden of airway inflammation in COPD has multiple potential applications in research and clinical practice. For example, clinical trials demonstrating that COPD patients with higher sputum eosinophil counts have a greater response to corticosteroids have led to the development of blood eosinophil counts as a biomarker to help direct inhaled corticosteroid use in clinical practice [4]. 

A biomarker is a quantifiable characteristic of a biological mechanism [6], and may also reflect the disease process(es) [7]. The use of validated biomarker methodology, with acceptable reproducibility, is crucial to potential applications in clinical trials, research studies and clinical practice [2,3,5]. For clinical trials, European and USA regulators (the European Medicines Agency and the U.S department of Health and Human Services Food and Drug Administration, respectively) have published recommendations for bioanalytical method validation of drug and/or metabolite assays [8,9]. Many of the principles of these recommendations can also be applied to ensure high quality validation of endogenous biomarker assays [10,11,12,13,14]. 

Sputum sampling is commonly used to measure airway inflammation in COPD studies, as it allows the measurement of inflammatory cells and mediators [15,16,17]. It is well accepted that the measurement of sputum supernatant biomarkers requires a high degree of validation, as many of the immunoassays used were not developed for this complex matrix [5]. Various publications have described validation activities for immunoassays in sputum supernatant [16,17,18,19,20,21,22]. However, many of these previous studies did not perform comprehensive assay validation according to regulatory standards, including analyte recovery during sample dilution, suitability of the reference standard, precision and reproducibility. The main novelty of the work described in this manuscript is the use of regulatory standards to assess sputum supernatant assay validity in COPD.

Our aim was to validate sputum supernatant biomarkers according to current regulatory standards and demonstrate their practical utility in COPD clinical studies. We describe first a comprehensive set of validation experiments on candidate biomarkers (part 1), followed by the subsequent practical application of validated biomarkers in clinical studies with COPD patients (part 2). This experimental strategy enabled the identification of validated sputum biomarkers that also have clinical relevance in COPD. The clinical evaluation included differences between COPD patients and controls and changes during COPD exacerbations. As COPD is a heterogeneous condition, we also evaluated differences between subgroups, classified according to smoking status, sputum cell counts and the presence of bacterial infection. The sputum supernatant volumes obtained from clinical sampling are often small and insufficient to perform multiple immunoassays, so different COPD cohorts were required to obtain sufficient supernatant to evaluate different immunoassays. 

## 2. Materials and Methods

### 2.1. Subjects

In part 1, sputum samples collected from healthy subjects and COPD patients were used for assay development. Part 2 included two cohorts who provided sputum samples during the stable state; cohort A comprised COPD patients (*n* = 30), healthy smokers (HS; *n* = 10) and healthy non-smokers (HNS; *n* = 10), while cohort B comprised of a different cohort of COPD patients (*n* = 81), HS (*n* = 15) and HNS (*n* = 26). In cohort B, we included sputum bacteriology and cell count data from 58 COPD patients and 9 HNS [23] and cell count data only from 54 COPD patients [24] which has previously been published. HS and HNS had normal lung function with no history of respiratory disease. HS had a smoking history of >10 pack years, and HNS < 1 pack year. All COPD patients had a smoking history of >10 pack years, and a physician diagnosis of COPD according to Global Initiative for Obstructive Lung Disease (GOLD) criteria, including a post bronchodilator first second of forced expiration/forced vital capacity (FEV_1_/FVC) ratio of <0.7. A subgroup of COPD patients (*n* = 14) in cohort A also provided a sputum sample during an exacerbation (Appendix A). All subjects provided written informed consent using protocols approved by the local Ethics committee (05/Q1402/41, 10/H1003/108 and 16/NW/0836). 

### 2.2. Sputum Processing

Induced sputum was processed as previously described [25]; details are included in the Appendix A. The supernatant was stored at −80 °C for later analysis. 

Cytospin preparations (Cytospin 4, Shandon, Runcorn, UK) were stained with Rapi-Diff II (Atom Scientific, Hyde, UK). In this case, 400 non-squamous cells were counted, and differential cell counts (DCC) obtained as a percentage of non-squamous cells. Cell viability was analysed by trypan blue exclusion. A subgroup of cohort B and HNS samples (*n* = 49 and 10, respectively) were processed for real time quantitative polymerase chain reaction (qPCR) detection of absolute abundance for *H.influenzae* (HI), *M. catarrhalis* (MC), *S. pneumoniae* (SP) and *Pseudomonas aeruginosa* (PA). Prior to the processing method outlined above, these samples were homogenised with PBS and glass beads in order to provide a sample for qPCR. Sputum supernatant samples were thawed directly on the day of analysis.

### 2.3. Part 1—Assay Development and Validation

#### 2.3.1. Assay Kits and Analysis

Assay kits were provided by commercial vendors (Appendix A). Where possible single lot numbers were used across assay development and validation. A total of 32 analyte measurements were assessed, quantified using 2 individual Enzyme linked immunosorbent assay (ELISA) assays, a 3-Plex Luminex assay and a 27-Plex Luminex assay. ELISA assays were analysed using a BioTek ELx808 plate reader (Swindon, UK) and Luminex assays were analysed using a Luminex MAGPIX platform (Austin, TX, USA). Samples were analysed in duplicate throughout assay development and validation. 

#### 2.3.2. Method Development

Method Development for all assays comprised of the following assessments:

##### Matrix Dilution

Initial method development runs were used to demonstrate that the endogenous analyte interacts with assay reagents in a similar manner to the recombinant reference material by investigating whether sample matrix and recombinant standard dilution curves demonstrated parallelism [26]. This was assessed using ≥5 sputum supernatants as previously described [11,12,27]; see Appendix A. The smallest dilution factor able to achieve parallelism was defined as the assay minimum required dilution (MRD) [27]. 

All matrix dilution assessments utilised assay specific diluents; see Appendix A. Analyte concentrations in dilutions subsequent to the MRD were acceptable if within 70–130% of the expected concentration calculated from the MRD sample [12]; calculation details are in the Appendix A. 

##### Standard Recovery

Spike recovery experiments were performed using the recombinant standard reference material. This is known as assay selectivity [11], which assesses the ability of the assay to measure the analyte in the presence of other matrix components. Selectivity was assessed using ≥5 sputum supernatants prepared at the MRD and ‘spiked’ with a known concentration of recombinant analyte, with the target total analyte concentration being within the standard curve range. The spiking solution represented less than 5% of the total sample volume. The previously described subtraction method was used to calculate percentage recovery (see Appendix A) [11,22]. Optimal recovery was within 70–130%. 

##### Establishment of Calibration Curve

A minimum of 10 standards of varying concentrations were prepared for each assay. Calibration curves were established over a minimum of 3 independent assay runs, to assess the precision and accuracy. All standards were prepared from a lyophilised reference standard material provided as part of the assay kits. 

Regression models were selected in the first method development run based on the co-efficient of determination (r^2^) and analysis of the percentage recovery demonstrated in the back-calculated concentration of each standard against the respective nominal concentration in order to select the regression model resulting in the lowest concentration lower limit of quantification (LLOQ) [28]. Over the subsequent runs, validated standards were required to demonstrate a Coefficient of variation (CV) of ≤20% (≤25% at the upper limit of quantification (ULOQ) and LLOQ) and recovery of ±20% (±25% at the ULOQ and LLOQ). 

#### 2.3.3. Method Validation

Method Validation for all assays comprised of the assessment of the following parameters:

##### Intra-Assay Precision

Six replicates of a sample were analysed in order to calculate the intra-assay precision CV% for samples at low, mid and high points of the assay quantitative range. This was repeated in three independent experiments using each sample, and the average CV% derived from the three assay runs defined the intra-assay precision. 

##### Inter-Assay Precision 

Inter-assay precision was assessed from repeated analysis of the same samples in duplicate across multiple independent runs (always ≥3 runs). Inter-assay precision (CV%) was calculated using the average concentration from each run of samples at the low, mid and high points of the assay quantitative range. 

##### Assay Limits of Quantification and Detection

The LLOQ is the lowest concentration of analyte that can be quantified with acceptable accuracy and precision. Initial estimation of LLOQ was performed during parallelism assessment. Where possible, the sample displaying the lowest concentration in the parallel dilution series was selected [26], with ≥6 replicates of this selected sample used in each validation run. The average concentration of the LLOQ sample calculated via collation of validation batch data defined the LLOQ of the assay. In cases where this was not possible, the lowest validated standard calibrator confirmed from collation of three independent standard curve preparations defined the assay LLOQ. 

The assay limit of detection (LOD) is the lowest amount of analyte that can be detected but not necessarily accurately quantified. LLOQs were required to be above the respective assay LOD, see Appendix A. 

##### Validated Analyte Assays 

The criteria that we used to classify an assay as validated were as follows; the matrix dilution, standard recovery and calibration curve acceptance criteria previously stated should be met. In cases where standard recovery was sub-optimal, inter-assay precision was required to be high (CV < 20%) in order to be classified as validated. 

### 2.4. Part 2—COPD versus Controls, and COPD Exacerbations

#### 2.4.1. Study Design

Spirometry was performed at all visits according to guidelines [29,30] (using the EasyOn PC Sensor, NDD, Intermedical). Sputum samples were collected during stable state, defined as no symptom-defined exacerbation or respiratory illness within 4 weeks of sampling. COPD patients were asked to contact the investigators as soon as possible following the onset of exacerbation symptoms, and where possible they were assessed by a physician within 24 h. Symptoms were assessed using COPD Assessment Test (CAT) and Modified Medical Research Council Questionnaire (mMRC) scores and health related quality of life using the St George’s Respiratory Questionnaire (SGRQ-C) [31]. 

#### 2.4.2. Quantitative PCR Detection of Common Respiratory Pathogens

Quantification of *H. influenzae* (HI)*, M. catarrhalis* (MC)*, S. pneumoniae* (SP) *and P. aeruginosa* (PA) was performed as per Beech et al. [23] and described in the Appendix A.

#### 2.4.3. Sputum Supernatant Biomarkers

Cohort A sputum supernatants were analysed for myeloperoxidase (MPO) and IL-8 by ELISA (R&D Systems, Abingdon, UK), along with interleukin (IL)-1β, IL-6 and Tumour necrosis factor (TNF)-α by 3-plex Luminex multiplex Assay (Merck Millipore, MA, USA). Assay quantitative ranges are available in Appendix A. 

Cohort B sputum supernatants were analysed for IL-1b, IL-1 receptor antagonist (RA), IL-2, IL-4, IL-5, IL-6, IL-7, IL-8, IL-9, IL-10, IL-12p70, IL-13, IL-15, IL-17A, Eotaxin, Basic fibroblast growth factor (FGF), Granulocyte-colony stimulating factor (G-CSF), Granulocyte-macrophage colony-stimulating factor (GM-CSF), Interferon (IFN)-γ, Interferon gamma-induced protein (IP)-10, Monocyte chemoattractant protein (MCP)-1, Macrophage inflammatory protein (MIP)-1α, MIP-1β, Platelet derived growth factor (PDGF)-BB, RANTES, TNF-α and Vascular endothelial growth factor (VEGF) by 27-plex Luminex multiplex assay (Bio-Rad, Hertfordshire, UK). Lot specific assay quantitative ranges are available in Appendix A. 

All immunoassays were performed in duplicate. Values below the assay LLOQ were reported as half the value of the LLOQ. The mean was used for statistical analysis. 

#### 2.4.4. Statistical Analysis

Comparisons of clinical characteristics between groups were performed using Fisher’s exact test, one-way ANOVA with Bonferroni post hoc test or the Krushkal Wallis test with Dunns post hoc test (depending on normality of data). COPD stable and exacerbation samples were analysed using paired *t*-tests or Wilcoxon signed rank tests. Pearsons correlation coefficient or Spearman’s Rank test were used to assess associations. Comparisons between COPD current and ex-smokers were performed using Mann-Whitney tests. *p* < 0.05 was considered statistically significant. All analyses were performed using GraphPad Prism version 9.1.2 (San Diego, CA, USA). 

## 3. Results

### 3.1. Part 1—Assay Development and Validation

#### 3.1.1. Method Development

##### Matrix Dilution

Assay specific MRDs were defined following matrix dilution assessments. The MRDs were set at 1:200, 1:5, 1:2 and 1:8 for the MPO ELISA, IL-8 ELISA, 3-plex assay and the 27-plex assay, respectively. Further details regarding MRD calculations are in the Appendix A. 26 of the 32 analytes were within the required acceptance criteria, hence demonstrating little matrix interference at MRD. Basic FGF, GM-CSF, IL-7, IL-9, IL-12p70 and PDGF-BB failed to meet the acceptance criteria (Table 1). Average %CV of samples analysed in duplicate at the MRD and subsequent dilutions was <17% for all analytes (Appendix A). 

Establishment of calibration curve data (Standard accuracy) is presented as the average %RE of validated standard calibrators from data collated from 3 independent runs. 

Intra-assay data presented as the average %CV of 6 replicates per run of a minimum of 2 samples at different points on the standard curve, across 3 individual runs. Inter-assay data presented as the average %CV of a minimum of 2 samples at different points on the standard curve, from data collated from 3 runs. 

##### Standard Recovery

26 and 25 (of 32 analytes) met acceptance criteria for recombinant analyte spikes at high and low points of the calibration curve, respectively. Analytes within the 27-plex that failed at either point were Eotaxin, GM-CSF, IL-1RA, IL-8, IL-9, IL-15, MIP-1β and VEGF. IL-1β, IL-6 and TNF-α failed to meet the acceptance criteria within the 3-plex assay but demonstrated acceptable recovery within the 27-plex assay. (Table 1). 

##### Establishment of Calibration Curve

Standard average %CV was <14% and average recovery was 96–104% for all analytes (Appendix A and Table 1). PDGF-BB and VEGF did not demonstrate a minimum of 6 validated standards. Further details are in the Appendix A.

#### 3.1.2. Method Validation

##### Precision

The average %CV within each assay (intra-assay precision) derived from a minimum of three independent assay runs was <12% for all analytes (Table 1). 

The average %CV derived from data collated from three independent assay runs (inter-assay precision) demonstrated a high level of precision (CV < 20%) for the majority of analytes (27/30 analytes; Table 1). 

##### Assay Limits of Quantification and Detection

LLOQs and LODs were defined for all analytes. LLOQ average %CV was less than 25% across all analytes within each assay (Appendix A). IL-1β, IL-6 and IL-8 within the 27-Plex assay had lower LLOQs compared to the 3-Plex assay and IL-8 ELISA (0.06 versus 0.89, 0.35 versus 0.48 and 2.51 versus 34.22 pg/mL, respectively). TNF-α had a higher LLOQ within the 27-Plex assay versus the 3-Plex assay (2.70 versus 0.970 pg/mL). 

##### Validated Analyte Measurements

A total of 25 analyte assessments were classed as validated. 6 analytes were rejected due to sub-optimal matrix dilution (Basic FGF, GM-CSF, IL-7, IL-9, IL-12p70 and PDGF-BB) and VEGF due to <6 validated standards. Of the 25 acceptable assessments, 8 analytes (Eotaxin, IL-1RA, IL-8, IL-15 and MIP-1β within the 27-plex and all 3-plex analytes) displayed sub-optimal standard recovery but inter-assay precision was <19% (for all analytes), thus meeting the acceptance criteria. For part 2, we focused on validated assays (unvalidated assay data is shown in the Appendix A). 

### 3.2. Part 2—COPD versus Controls and COPD Exacerbations

#### 3.2.1. Cohort A

The baseline characteristics of 30 COPD patients, 10 HS and 10 HNS are presented in Table 2. COPD patients were significantly older than HS and HNS (67.7 versus 59.4 and 53.6 years, *p* = 0.01 and <0.01, respectively). COPD patients had a historical mean exacerbation rate/year of 1.8, while mean CAT and total SGRQ score were 22.5 and 57.2, respectively.

Sputum neutrophil % was numerically higher in COPD patients compared to HS and HNS, without reaching statistical significance (83.7% versus 69.1% (*p* = 0.14) and 66.4% (*p* = 0.07), respectively). Sputum eosinophil % and absolute counts were significantly higher in COPD patients compared to HNS (*p* ≤ 0.01). Macrophage % was significantly lower in COPD patients compared to both HS and HNS (*p* = 0.02 and <0.01, respectively). 

In the stable state, IL-6, TNF-α and IL-8 were higher in COPD patients compared to both HS and HNS (*p* < 0.05 for all comparisons, Figure 1). IL-1β was higher in some COPD patients, reaching significance versus HS (*p* = 0.04) but not HNS (*p* = 0.85). MPO was numerically increased in COPD patients compared to HS and HNS, without statistical significance (*p* = 0.06 and 0.33, respectively, Figure 1). 8 COPD samples >ULOQ at initial analysis (analysed at 1:200) were reanalysed at 1:1600 but remained >ULOQ. No further sputum matrix was available for re-analysis at a higher dilution factor and so they were assigned the highest measurable MPO concentration for samples analysed at 1:1600 (16,000 ng/mL). 

COPD sputum samples collected during exacerbations (*n* = 14) show an increase in neutrophil percentage (82.8% versus 92%, *p* < 0.05, Appendix A), IL-1β, IL-6 and TNF-α compared to baseline (*p* < 0.05 for all comparisons; Figure 2). MPO and IL-8 levels did not change during exacerbations (*p* > 0.05). 

#### 3.2.2. Cohort B

The baseline characteristics for 81 COPD patients, 15 HS and 26 HNS are presented in Table 3. COPD patients were significantly older than HS and HNS (*p* = 0.04 and <0.01, respectively). In the COPD group, the mean FEV_1_ % predicted was 64.7%, while mean CAT and total SGRQ scores were 19.7 and 50.3, respectively. The historical mean exacerbation rate/year was 1.0. 

Sputum neutrophil (%) was similar in COPD patients compared to both HS and HNS (68.75% versus 72.25 and 68.38, respectively). Sputum eosinophil % and absolute counts were higher in COPD patients versus HNS (*p* < 0.01 for both comparisons), while absolute eosinophil counts were higher in COPD compared to HS (*p* = 0.04) (Table 3). 

Using the 27 plex, IL-6 and IL-8 levels were significantly increased in COPD patients compared to HS and HNS (*p* < 0.05 for all comparisons). No other analytes differed between groups. (Table 4). 

48 COPD patients were ex-smokers. Sputum neutrophil % was significantly higher in COPD ex-smokers compared to current smokers; 64.9% versus 76.8%, respectively, *p* = 0.005. IL-1β, IL-4, IL-8, G-CSF, IFN-γ, IP-10, MCP-1, MIP-1α, MIP-1β and TNF-α were also significantly higher in ex-smokers (p < 0.01 for all comparisons, Figure 3). Inhaled Corticosteroid (ICS) use had no effect on sputum cytokine levels (Appendix A). 

For all subjects combined (COPD patients and controls), IL-1β, IL-1RA, IL-2, IL-4, IL-6, IL-8, IL-17A, G-CSF, IFN-γ, IP-10, MIP-1α, MIP-1β and TNF-α were significantly correlated with sputum neutrophil percentage (Table 5). All of these, with the exception of IL-1RA and IL-6, remained significantly correlated when analysed in COPD patients. 

49 COPD patients and 10 HNS had sputum bacterial qPCR data; 15 COPD patients had HI levels above the HNS range [23]. Far fewer patients were above the HNS range for SP (*n* = 2) and MC (*n* = 2), and none for PA; these SP and MC colonised patients also had evidence of HI co-infection (details in Appendix A). Sputum neutrophil % was significantly higher in HI^+ve^ COPD compared to HI^−ve^ and HNS (*p* = 0.02 and 0.03, respectively, Figure 4). IL-6 was significantly increased in both HI^+ve^ and HI^−ve^ COPD patients compared to HNS (Figure 5). IL-4, IL-8, and G-CSF were significantly increased in HI^+ve^ COPD patients compared to HI^−ve^ and HNS. IL-1β and IFN-γ were significantly increased in the HI^+ve^ group compared to the HI^−ve^ group, with IP-10 and TNF-α trending towards significance (*p* = 0.07 and 0.05, respectively, Appendix A). IL-1β was significantly increased in HNS compared to HI^−ve^ COPD. 

## 4. Discussion

A comprehensive development and validation process resulted in validation of the majority of analyte measurements in sputum samples. Of note, the characteristics of the 3-plex assay were different to the same analytes measured by 27-plex assay, including standard recovery and LLOQs, although all assays met acceptance criteria. The immunoassays demonstrated some differences in COPD patients compared to controls. The results for the common cytokines studied (IL-1β, IL-6, TNF-α and IL-8) were not identical across cohorts, which can be explained by both differences between immunoassays and differences in cohort characteristics. Nevertheless, an interesting pattern emerged of cytokine expression profiles associated with current smoking effects, sputum neutrophilia and the presence of bacterial infection in COPD patients. 

Two key components of assay validation are matrix dilution, which compares measurements using the recombinant reference material and the endogenous biomarker across a range of dilutions, and standard recovery, which assesses the ability of the assay to measure known concentrations of the endogenous biomarker [32]. Of 32 analytes studied, 6 and 11 analytes did not meet acceptance criteria for matrix dilution and standard recovery, respectively. Standard recovery in the 3-plex assay did not meet acceptance criteria, while the same analytes within the 27-plex assay met acceptance criteria. Despite this limitation, the 3-plex assay showed acceptable intra- and inter-assay precision, indicating that suboptimal standard recovery was not associated with excessive assay variability. Furthermore, the 3-plex assay demonstrated differences in COPD patients versus controls, and cytokine upregulation during exacerbations. Sub-optimal standard recovery may result in measured levels that are lower than the true values, meaning that the absolute results from such an immunoassay cannot be directly compared to other immunoassay data. However, we have demonstrated that such assays (with sub-optimal standard recovery) may still be useful if used consistently within a study population. 

The LLOQ assessment demonstrated that IL-1β sensitivity was greater for the 27-Plex versus 3-plex assay. This resulted in 48% versus 3% of IL-1β measurements being undetectable in cohort A compared to B, respectively. The sub-optimal 3-plex assay standard recovery may lead to reported results being are falsely low, which can further reduce the 3-Plex assay ability to detect IL-1β at low concentrations. Reduced assay sensitivity (LLOQ) combined with reduced standard recovery likely contributed to more undetectable levels in cohort A, and therefore differences in the results observed between cohorts. 

Ideally, the comparison of immunoassays in part 2 would be conducted on a single cohort of COPD patients, but the limited supernatant volumes commonly obtained from sputum sampling meant that we needed two cohorts to evaluate the different immunoassays. COPD sputum neutrophil percentage counts were higher in cohort A versus B (83.7% and 68.8%, respectively). Some of the 3-plex cytokines (IL-1β and TNF-α) were positively correlated with neutrophil percentages in Cohort B, so similar neutrophil counts between COPD patients and controls in cohort B reduces the possibility to elucidate between group differences for these cytokines. Previous studies have shown conflicting results for IL-1β and TNF-α, with increased levels in COPD patients and no difference versus controls being reported [33,34]. Nevertheless, despite the differences in clinical and immunoassay characteristics between cohorts, there were significant differences between COPD patients and controls for IL-6 and IL-8 in both cohorts, consistent with previous studies [15,33,34,35]. We also showed statistically significant increases in IL-1β, IL-6 and TNF-α during an exacerbation, as previously reported [36,37]

Matrix dilution can demonstrate (by parallelism) that endogenous and recombinant proteins have similar immunoaffinity to the assay reagents. We therefore classified assays that did not meet parallelism criteria as unvalidated. Interestingly, 8 analytes met matrix dilution assessment acceptance criteria but demonstrated sub-optimal standard recovery. This is indicative of a specific interference present in the endogenous matrix that affects the measurement of the recombinant reference material and the endogenous biomarker in a similar way. In these cases, we required a high level of inter-assay precision (as already discussed) for assays to be classified as validated. Some studies have used standard recovery alone to validate sputum immunoassays [16,17]. We took a more comprehensive approach, using a wide range of assay criteria to determine assay validity. We demonstrate the importance of the assessment of both matrix dilution and standard recovery during assay validation as assessment of standard recovery alone may result in rejection of potentially useful assays. 

The need for sputum immunoassay validation was recognised over 20 years ago [20]. The European Respiratory Society (ERS) published recommendations for sputum soluble mediator measurement validation [38]. Recommendations included standard recovery assessment using sputum plugs spiked prior to processing to evaluate matrix dilution and standard recovery. These ERS recommendations appeared to influence subsequent publications, [16,39], as standard recovery assessment from spiked sputum plugs was used. Spiking sputum plugs incorporates an assessment of whether sputum processing methodology allows adequate retrieval of endogenous analytes. Our approach focused more on immunoassay performance, accepting that the degree of retrieval/loss during processing is an important but separate question [20]. 

Sputum neutrophil percentage has previously been shown to be increased in COPD ex-smokers versus current smokers [24]. A potential explanation is that acute exposure to cigarette smoke extract down regulates the gene expression of multiple pro-inflammatory cytokines [40,41]. Higher sputum neutrophil percentage and cytokine expression following smoking cessation may be due to a reduction in acute smoking induced cell death and immunosuppressant effects [42]. We also observed reduced sputum neutrophil counts and inflammatory cytokine levels in COPD current smokers, with 9 out of the 10 cytokines higher in COPD ex-smokers positively correlated with sputum neutrophil percentage. 

In Cohort B, we used previously defined bacterial colonisation thresholds in healthy subjects to categorise stable state COPD patients into HI^+ve^ and HI^−ve^ groups [23]. IL-1β, and IFN-γ were significantly increased in the HI^+ve^ group compared to HI^−ve^, with IL-4, IL-8 and G-CSF being higher versus both HI^−ve^ and HNS. Previous studies have reported higher IL-1β and TNF-α levels in HI^+ve^ versus HI^−ve^ COPD patients [43,44]; we show similar results with an increase in TNF-α within the HI^+ve^ group tending towards statistical significance (*p* = 0.05). Notably, all of the cytokines increased within the COPD HI^+ve^ group were positively correlated with sputum neutrophil percentage. The presence of H. influenzae colonisation is known to be associated with increased neutrophilic inflammation [43,45,46], sputum neutrophil percentage was significantly increased in the HI^+ve^ group compared to HI^−ve^ and HNS within our study and now we demonstrate cytokines profiles associated with H. influenzae colonisation and neutrophilia. Wang et al. [43] reported two neutrophilic COPD subgroups, differentiated by H. influenzae predominance in sputum. Wang et al. also reported that sputum IL-6 level were not associated with H. influenzae colonisation, supported by our findings [43]. Winslow et al. [47] described an IL-6 trans-signalling (IL-6TS) high COPD subset, defined by elevated levels of sputum IL-1β, IL-6, soluble IL-6 receptor, MIP-1β and IL-8, with increased sputum neutrophilia and HI colonisation. We also report associations between sputum IL-1β, IL-8 and MIP-1β with sputum neutrophils, and IL-1β and IL-8 with H. influenzae colonisation. 

Our findings highlight the importance of sputum immunoassay validation prior to application within a clinical study. The validation results for immunoassays that measure the same analyte can differ, which may influence the results obtained when comparing COPD patients to controls. Despite these limitations, the application of a validated 27-plex assay demonstrated associations between H. influenzae colonisation and higher levels of selected cytokines, and the effects of current smoking. We conclude that immunoassay validation enabled inflammatory mediators associated with different COPD characteristics to be determined

## Figures and Tables

**Figure 1 biomedicines-10-01949-f001:**
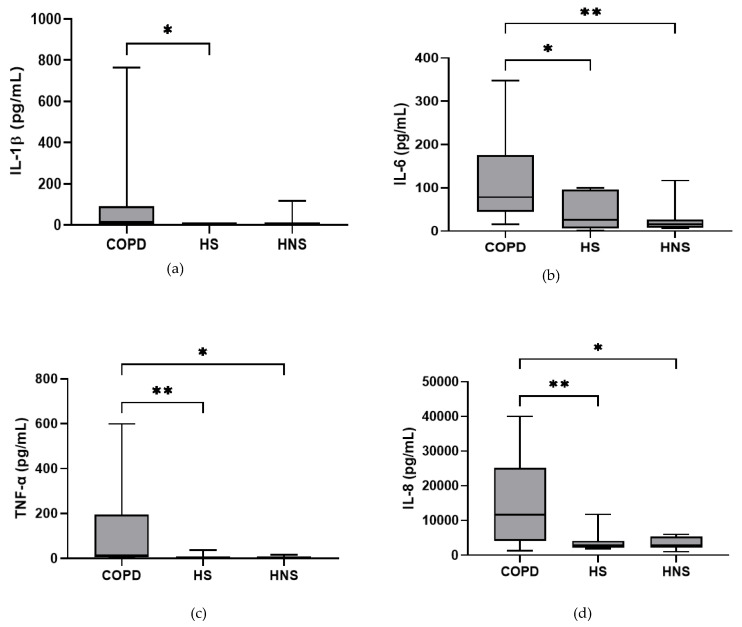
**Sputum Biomarkers in chronic obstructive pulmonary disease (COPD) (*n* = 30), healthy smokers (HS) (*n* = 10) and healthy non-smokers (HNS) (*n* = 10):** (**a**) Interleukin (IL-)1β, data is presented as minimum, maximum and median concentrations, data analysed using Krushkal-Wallis test with Dunns post hoc test. * *p* ≤ 0.05. (**b**) IL-6, data is presented as minimum, maximum and median concentrations, data analysed using Krushkal-Wallis test with Dunns post hoc test. * *p* < 0.05, ** *p* < 0.01. (**c**) Tumour necrosis factor (TNF)-α, data is presented as minimum, maximum and median concentrations, data analysed using Krushkal-Wallis test with Dunns post hoc test. * *p* < 0.05, ** *p* < 0.01. (**d**) IL-8, data is presented as minimum, maximum and median concentrations, data analysed using Krushkal-Wallis test with Dunns post hoc test. * *p* < 0.05, ** *p* < 0.01. (**e**) Myeloperoxidase (MPO), data is presented as minimum, maximum and median concentrations, data analysed using Krushkal-Wallis test with Dunns post hoc test.

**Figure 2 biomedicines-10-01949-f002:**
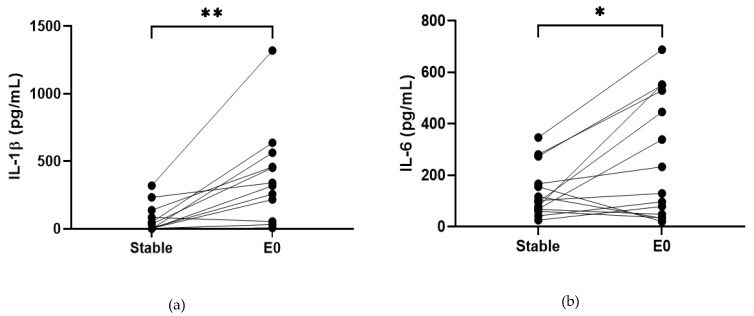
**Sputum Biomarkers in COPD Stable and Exacerbating Samples**: (**a**) IL-1β, data is presented as paired concentrations, data analysed using Wilcoxon signed rank test. ** *p* ≤ 0.01. (**b**) IL-6, data is presented as paired concentrations, data analysed using a paired *t*-test. * *p* < 0.05. (**c**) TNF-α, data is presented as paired concentrations, data analysed using Wilcoxon signed rank test. ** *p* < 0.01. (**d**) IL-8, data is presented as paired concentrations. (**e**) MPO, data is presented as paired concentrations.

**Figure 3 biomedicines-10-01949-f003:**
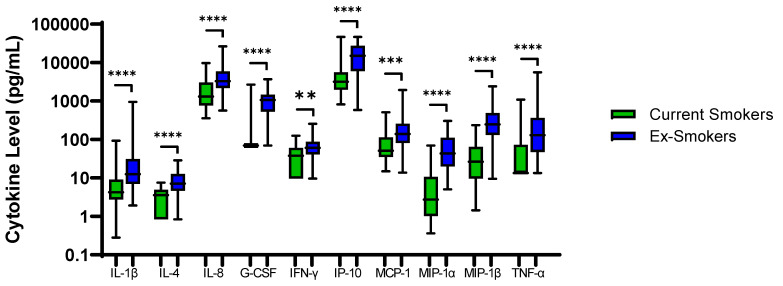
**Sputum Biomarkers in COPD Ex-Smokers and COPD Current Smokers**: Data is presented as minimum, maximum and median concentrations, data analysed using Mann Whitney test. ** *p* ≤ 0.01, *** *p* ≤ 0.001 **** *p* ≤ 0.0001.

**Figure 4 biomedicines-10-01949-f004:**
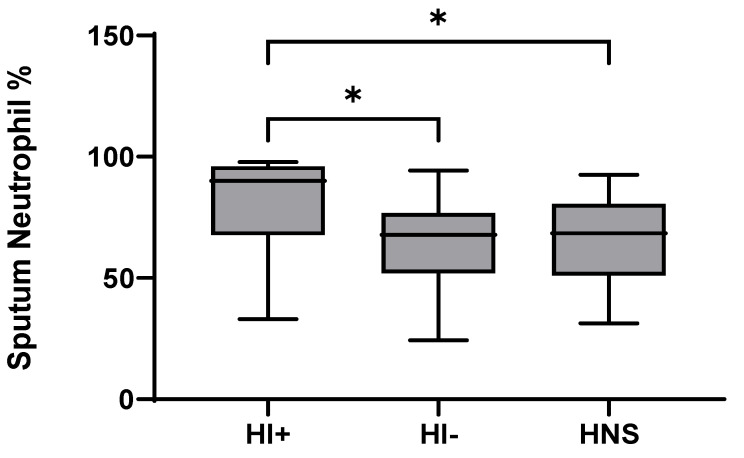
**Sputum Neutrophil Percentage in COPD HI^+ve,^ COPD HI−^ve^ and HNS**: Data is presented as minimum, maximum and median concentrations, data analysed using Krushkal-Wallis test with Dunns post hoc test. * *p* ≤ 0.05.

**Figure 5 biomedicines-10-01949-f005:**
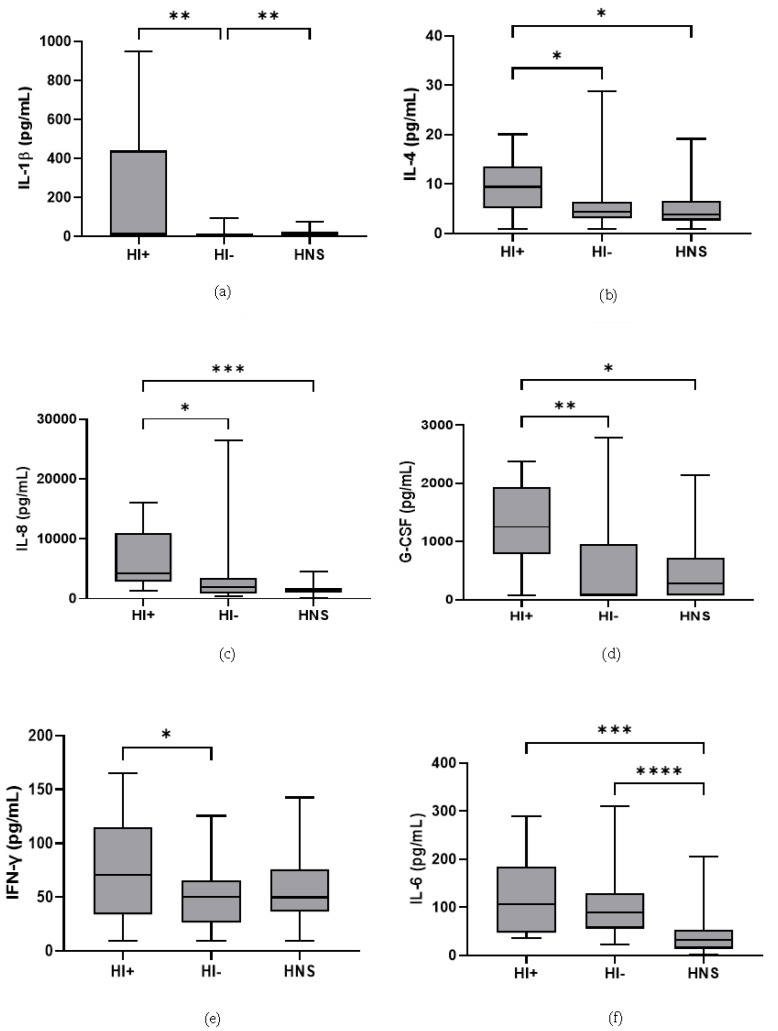
**Sputum Biomarkers in COPD HI^+ve,^ COPD HI^−ve^ and HNS**: (**a**) IL-1β, data is presented as minimum, maximum and median concentrations, data analysed using Krushkal-Wallis test with Dunns post hoc test. ** *p* ≤ 0.01. (**b**) IL-4, data is presented minimum, maximum and median concentrations, data analysed using Krushkal-Wallis test with Dunns post hoc test. * *p* ≤ 0.05. (**c**) IL-8, data is presented as minimum, maximum and median concentrations, data analysed using Krushkal-Wallis test with Dunns post hoc test. * *p* ≤ 0.05, *** *p* ≤ 0.001. (**d**) Granulocyte colony stimulating factor (G-CSF), data is presented as minimum, maximum and median concentrations, data analysed using Krushkal-Wallis test with Dunns post hoc test. * *p* ≤ 0.05, ** *p* ≤ 0.01. (**e**) Interferon (IFN)-γ, data is presented as minimum, maximum and mean concentrations, data analysed using a one-way ANOVA test with Bonferroni post hoc test. * *p* ≤ 0.05. (**f**) IL-6, data is presented as minimum, maximum and median concentrations, data analysed using Krushkal-Wallis test with Dunns post hoc test. *** *p* ≤ 0.001, **** *p* ≤ 0.0001.

**Table 1 biomedicines-10-01949-t001:** Method Development and Validation Results.

Analyte	Matrix Dilution (%RE)	High Standard Recovery (%RE)	Low Standard Recovery (%RE)	Standard Accuracy (%RE)	Intra-Assay (%CV)	Inter-Assay (%CV)
**ELISA Assays**						
**MPO**	102.55	110.64	83.00	100.00	2.55	3.31
**IL-8**	99.70	92.10	101.84	102.22	7.04	7.82
**3-Plex Luminex Assay**						
**IL-1β**	75.44	36.07	35.47	100.43	11.12	11.55
**IL-6**	114.96	65.30	111.64	100.46	8.53	10.24
**TNF-α**	124.44	34.24	17.95	100.27	11.82	8.75
**27-Plex Luminex Assay**						
**Basic FGF**	144.32	124.60	80.86	100.03	9.28	9.52
**Eotaxin**	97.83	98.10	54.39	100.61	5.35	18.59
**G-CSF**	103.06	75.74	106.30	101.57	5.66	5.83
**GM-CSF**	144.12	89.63	60.49	101.24	2.52	1.87
**IFN-y**	117.40	73.90	77.12	99.61	5.60	24.44
**IL-1β**	105.19	73.91	82.76	101.23	5.36	7.67
**IL-1RA**	116.92	66.94	94.04	100.85	3.09	4.86
**IL-2**	104.22	84.24	90.45	100.21	4.42	6.06
**IL-4**	110.14	75.07	81.33	99.05	6.58	12.73
**IL-5**	103.63	76.38	74.25	99.96	3.19	3.61
**IL-6**	95.58	90.26	79.49	103.41	4.15	9.58
**IL-7**	137.696	97.59	87.54	100.47	5.09	23.44
**IL-8**	111.87	76.93	−97.11	100.66	3.97	5.36
**IL-9**	149.10	65.22	74.11	100.37	7.28	11.02
**IL-10**	<LLOQ	71.34	71.17	100.63	3.80	6.22
**IL-12p70**	146.01	89.68	84.09	100.03	3.24	4.61
**IL-13**	<LLOQ	82.77	74.84	103.24	3.16	6.09
**IL-15**	118.72	75.86	57.99	100.82	2.95	2.80
**IL-17A**	108.64	85.46	90.16	100.11	2.60	2.62
**IP-10**	87.38	76.32	<LLOQ	99.66	3.84	36.17
**MCP-1**	112.38	92.57	80.56	102.90	8.00	7.17
**MIP-1a**	87.35	86.23	84.54	102.33	4.00	4.03
**MIP-1b**	107.42	87.33	25.16	100.86	3.39	10.85
**PDGF-BB**	130.83	120.17	70.97	<6	Unvalidated Assay	
**RANTES**	128.31	87.11	80.46	102.12	8.78	19.52
**TNF-α**	109.45	110.77	94.15	100.21	4.27	32.38
**VEGF**	94.08	48.82	85.30	<6	Unvalidated Assay	

*Matrix dilution data is presented as the average %RE derived from serial dilutions, with the minimum required dilution (MRD) acting as the reference matrix. Myeloperoxidase (MPO) analysed within samples prepared at 1:200 and 1:400 dilutions using kit provided RD5K diluent, n = 5. Interleukin (IL-)8 analysed within samples prepared at 1:5, 1:10. 1:20, 1:40 and 1:80 dilutions using proprietary diluent, n = 5. Luminex 3-Plex analysed within samples prepared at 1:2, 1:4. 1:8 and 1:16 dilutions using proprietary diluent, n = 9. Luminex 27-Plex analysed within samples prepared at 1:8, 1:16. 1:32 and 1:64 dilutions using proprietary diluent, n = 7. Standard recovery data is presented as average %RE of recombinant reference standard spiked into sputum matrix prepared at the assay MRD. Recombinant reference standard was spiked at low and high analyte concentrations relative to the appropriate standard curve. MPO analysed within samples prepared at 1:200 using kit provided RD5K diluent, n = 5. IL-8 analysed within samples prepared at 1:5 dilutions using proprietary diluent, n = 5. Luminex 3-Plex analysed within samples prepared at 1:2 using proprietary diluent, n = 2. Luminex 27-Plex analysed within samples prepared at 1:8 using proprietary diluent, n = 5.*

**Table 2 biomedicines-10-01949-t002:** Baseline characteristics for COPD subjects, healthy smoking and healthy non-smoking controls in Cohort A (*n* = 30, 10 and 10, respectively *).

Characteristics	COPD (*n* = 30)	HS (*n* = 10)	HNS (*n* = 10)	*p*-Value (COPD vs. HS)	*p*-Value (COPD vs. HNS)
**Gender (% Male)**	80.0	50.0	50.0	0.10	0.10
**Age**	67.7 (6.7)	59.4 (7.7)	53.6 (7.2)	0.01	<0.01
**Smoking Status (Current %)**	30.0	50.0	n/a	0.28	n/a
**Pack Years**	51.2 (22.0)	27.9 (11.1)	0.04 (0.1)	<0.01	<0.01
**BMI (kg/m^2^)**	26.9 (5.0)	24.5 (2.7)	28.1 (3.8)	>0.99	>0.99
**Retrospective Exacerbation Rate (1-year period)**	1.0 [0.0–11.0]	n/a	n/a	n/a	n/a
**0 (%)**	37.9	n/a	n/a	n/a	n/a
**1 (%)**	20.7	n/a	n/a	n/a	n/a
**≥2 (%)**	41.4	n/a	n/a	n/a	n/a
**Post FEV_1_ (L)**	1.5 (0.6)	2.7 (0.7)	3.1 (0.7)	<0.01	<0.01
**Post FEV_1_ (%)**	56.2 (19.6)	94.9 (8.7)	101.6 (12.5)	<0.01	<0.01
**Post FEV_1_/FVC Ratio (%)**	41.5 (12.1)	72.7 (3.8)	75.9 (3.5)	<0.01	<0.01
**Gold Category (%)**					
**1**	10.0	n/a	n/a	n/a	n/a
**2**	56.7	n/a	n/a	n/a	n/a
**3**	23.3	n/a	n/a	n/a	n/a
**4**	10.0	n/a	n/a	n/a	n/a
**CAT**	22.5 (7.2)	n/a	n/a	n/a	n/a
**mMRC**	3.0 [1.0–4.0]	n/a	n/a	n/a	n/a
**SGRQ-C (Total)**	57.2 (18.3)	n/a	n/a	n/a	n/a
**ICS Use (*n*)**	26 ^+^	n/a	n/a	n/a	n/a
**Sputum Characteristics**					
**Neutrophil (%)**	83.7 [24.5–99.8]	69.1 [38.3–86.8]	66.4 [55.0–82.0]	0.14	0.07
**Macrophage (%)**	9.9 [0.3–67.8]	28.4 [11.5–40.5]	29.6 [17.0–43.0]	0.02	<0.01
**Eosinophil (%)**	1.8 [0.0–13.2]	0.3 [0.0–33.3]	0.1 [0.0–2.5]	0.25	<0.01
**Lymphocyte (%)**	0.0 [0.0–2.0]	0.0 [0.0–0.5]	0.0 [0.0–0.8]	>0.99	>0.99
**Epithelial (%)**	2.2 [0.0–14.0]	0.9 [0.0–2.0]	0.9 [0.8–4.0]	0.11	0.57
**TCC × 10^6^/g**	7.9 [0.7–35.3]	6.2 [2.0–13.2]	6.8 [4.4–12.6]	0.86	>0.99
**Neutrophil cell × 10^6^/g**	7.1 [0.3–31.2]	3.4 [1.2—8.9]	5.3 [2.5–8.9]	0.56	>0.99
**Macrophage cell × 10^6^/g**	0.8 [0.0–7.1]	1.4 [0.3–4.0]	2.0 [0.9–4.4]	0.79	0.07
**Eosinophil cell × 10^6^/g**	0.2 [0.0–2.1]	0.0 [0.0–2.8]	0.0 [0.0–0.2]	0.19	0.01
**Lymphocyte cell × 10^6^/g**	0.0 [0.0–0.3]	0.0 [0.0–0.0]	0.0 [0.0–0.1]	>0.99	>0.99
**Epithelial cell × 10^6^/g**	0.1 [0.0–2.3]	0.0 [0.0–0.3]	0.1 [0.0–0.9]	0.05	>0.99

** The following data were not available for chronic obstructive pulmonary disease (COPD) subjects; 2 St George’s Respiratory Questionnaire (SGRQ-C), 2 sputum differential cell counts and 7 sputum absolute counts. ^+^ Further details on inhaled corticosteroid (ICS) use are available in Appendix A*
*. Data presented as mean (SD) or median [range] as appropriate. Healthy smoker (HS) and Healthy non-smoker (HNS) compared to COPD using ANOVA with Bonferroni post-hoc analysis or Kruskal-Wallis with Dunns post-hoc analysis or Kruskal-Wallis with Dunns post-hoc analysis as appropriate. Categorical data were compared between groups using a Fischer’s exact test. ns: non-significant. Spirometric measurement are post-bronchodilator values for COPD and pre-bronchodilator values reported for HS and HNS. TCC—Total cell count.*

**Table 3 biomedicines-10-01949-t003:** Baseline characteristics for COPD subjects, healthy smoking and healthy non-smoking controls in Cohort B (*n* = 81, 15 and 26, respectively *).

Characteristics	COPD (*n* = 81)	HS (*n* = 15)	HNS (*n* = 26)	*p*-Value (COPD vs HS)	*p*-Value (COPD vs HNS)
Gender (% Male)	59.3	46.7	53.8	0.40	0.65
Age	66.1 (7.3)	60.6 (7.7)	60.0 (9.4)	0.04	<0.01
Smoking Status (Current %)	40.7	36.4	0.0	0.39	n/a
Pack Years	42.6 (20.2)	25.2 (9.7)	0.0 (0.0)	<0.01	<0.01
BMI (kg/m^2^)	28.0 (5.3)	27.8 (3.7)	27.8 (3.5)	>0.99	>0.99
Retrospective Exacerbation Rate (1-year period)	1.0 [0.0–4.0]	n/a	n/a	n/a	n/a
0 (%)	40.7	n/a	n/a	n/a	n/a
1 (%)	34.6	n/a	n/a	n/a	n/a
≥2 (%)	24.7	n/a	n/a	n/a	n/a
Post FEV_1_ (L)	1.7 (0.5)	2.8 (0.5)	3.0 (1.0)	>0.01	>0.01
Post FEV_1_ (%)	64.7 (15.3)	103.4 (14.1)	106.3 (12.6)	>0.01	>0.01
Post FEV_1_/FVC Ratio (%)	51.9 (11.1)	75.0 (4.4)	76.1 (3.7)	>0.01	>0.01
Gold Category (%)					
1	17.3	n/a	n/a	n/a	n/a
2	65.4	n/a	n/a	n/a	n/a
3	17.3	n/a	n/a	n/a	n/a
4	0.0	n/a	n/a	n/a	n/a
CAT	19.7 (7.4)	n/a	n/a	n/a	n/a
mMRC	4.0 [0.0–4.0]	n/a	n/a	n/a	n/a
SGRQ-C (Total)	50.3 (17.5)	n/a	n/a	n/a	n/a
ICS Use (*n*)	56 ^+^	n/a	n/a	n/a	n/a
Sputum Characteristics					
Neutrophil (%)	68.75 [21.50–97.75]	72.25 [5.25–90.0]	68.38 [31.25–92.50]	>0.99	>0.99
Macrophage (%)	21.50 [1.00–72.25]	23.50 [5.75–86.75]	26.75 [5.00–57.50]	0.61	0.45
Eosinophil (%)	1.00 [0.00–16.50]	0.63 [0.00–2.00]	0.00 [0.00–3.50]	0.08	<0.01
Lymphocyte (%)	0.25 [0.00–3.50]	0.50 [0.00–1.25]	0.50 [0.00–3.75]	0.65	0.37
Epithelial (%)	2.50 [0.00–60.50]	1.88 [0.25–7.25]	3.00 [0.50–13.00]	0.74	>0.99
TCC × 10^6^/g	6.98 [0.62–100.9]	7.09 [1.48–17.36]	6.43 [0.99–32.18]	>0.99	>0.99
Neutrophil cell × 10^6^/g	4.43 [0.32–98.08]	5.00 [0.21–15.14]	4.96 [0.72–25.18]	>0.99	>0.99
Macrophage cell × 10^6^/g	1.24 [0.18–4.53]	1.85 [0.79–4.71]	2.06 [0.19–6.60]	0.38	0.46
Eosinophil cell × 10^6^/g	0.07 [0.00–1.29]	0.01 [0.00–0.12]	0.00 [0.00–0.44]	0.04	<0.01
Lymphocyte cell × 10^6^/g	0.02 [0.00–0.63]	0.03 [0.00–0.22]	0.03 [0.00–0.16]	0.76	>0.99
Epithelial cell × 10^6^/g	0.18 [0.00–3.86]	0.12 [0.02–0.51]	0.17 [0.03–0.75]	0.54	>0.99

** The following data were not available for COPD subjects; 1 COPD assessment test (CAT), 1 modified medical research council (mMRC), 15 SGRQ-C and 2 sputum differential and absolute cell counts. For HS subjects; 1 sputum differential and absolute cell count. For HNS subjects; 1 sputum absolute count. ^+^ Further details on ICS use are available in Appendix A*
*. Data presented as mean (SD) or median [range] as appropriate. HS and HNS compared to COPD using ANOVA with Bonferroni post-hoc analysis or Kruskal-Wallis with Dunns post-hoc analysis. Categorical data were compared between groups using a Fischer’s exact test. HS: healthy smoker; HNS: healthy non-smoker; ns: non-significant. ^a^ Spirometric measurement are post-bronchodilator values for COPD and pre-bronchodilator values reported for HS and HNS.*

**Table 4 biomedicines-10-01949-t004:** Sputum Supernatant Cytokines for COPD subjects, healthy smoking and healthy non-smoking controls in Cohort B (*n* = 81, 15 and 26, respectively).

Analyte	COPD (*n* = 81)	HS (*n* = 15)	HNS (*n* = 26)	*p*-Value (COPD vs HS)	*p*-Value (COPD vs HNS)
IL-1β (pg/mL)	8.92 [0.28–947.90]	13.46 [1.86–72.66]	14.52 [1.65–76.80]	0.61	0.16
IL-1RA (pg/mL)	7032 [2198–31,785]	6779 [2495–31,083]	8632 [3566–19,704]	>0.99	0.40
IL-2 (pg/mL)	6.80 [6.80–28.98]	6.80 [6.80–25.99]	6.80 [6.80–15.85]	>0.99	>0.99
IL-4 (pg/mL)	5.05 [0.84–28.73]	3.28 [0.84–22.12]	3.77 [0.84–19.14]	0.27	0.60
IL-6 (pg/mL)	85.98 [13.71–485.10]	40.09 [1.68–318.20]	33.21 [1.68–205.90]	0.01	<0.0001
IL-8 (pg/mL)	2863 [354.90–26,518]	1364 [269.60–5820]	1308 [152.40–4514]	0.02	0.003
IL-17A (pg/mL)	10.68 [10.69–58.92]	10.68 [10.68–65.38]	10.68 [10.68–63.14]	>0.99	>0.99
Eotaxin (pg/mL)	55.96 [4.73–253.3]	35.96 [14.87–194.40]	35.13 [5.34–151.90]	0.16	0.12
G-CSF (pg/mL)	497.30 [69.48–3710]	69.48 [69.48–1735]	284.40 [69.48–2137]	0.59	0.76
IFN-γ (pg/mL)	53.97 [9.60–256.60]	46.83 [9.60–143.10]	49.78 [9.60–142.40]	>0.99	>0.99
IP-10 (pg/mL)	6724 [587.90–46,318]	7026 [673–46,318]	6154 [721.60–46,318]	>0.99	>0.99
MCP-1 (pg/mL)	95.81 [13.69–1946]	47.22 [9.36–365]	61.32 [10.23–481.70]	0.09	0.08
MIP-1α (pg/mL)	20.20 [0.36–304]	11.96 [0.36–390.80]	23.21 [1.76–304.90]	0.91	>0.99
MIP-1β (pg/mL)	129.70 [1.44–2417]	69.09 [1.44–1126]	133.30 [16.31–967.20]	0.54	>0.99
TNF-α (pg/mL)	64.50 [13.40–5561]	57.88 [13.40–4581]	57.10 [13.40–401.60]	>0.99	>0.99

*Data presented as median [range]. HS and HNS compared to COPD using ANOVA with Bonferroni post-hoc analysis or Kruskal-Wallis with Dunns post-hoc analysis. HS: healthy smoker; HNS: healthy non-smoker; ns: non-significant. IL-5, IL-10, IL-13, IL-15, and RANTES were not detectable.*

**Table 5 biomedicines-10-01949-t005:** Cohort B: Sputum Neutrophil Percentage and Supernatant Cytokine Correlations.

Analyte	COPD, HNS, S (*n* = 119)	COPD (*n* = 79)
IL-1β (pg/mL)	rho = 0.4739, *p* < 0.0001	rho = 0.5136, *p* < 0.0001
IL-1RA (pg/mL)	rho = 0.2269, *p* = 0.01	rho = 0.1558, *p* = 0.17
IL-2 (pg/mL)	rho = 0.2703, *p* = 0.003	rho = 0.3399, *p* = 0.002
IL-4 (pg/mL)	rho = 0.2676, *p* = 0.003	rho = 0.3749, *p* = 0.0007
IL-6 (pg/mL)	rho = 0.1888, *p* = 0.04	rho = 0.1291, *p* = 0.26
IL-8 (pg/mL)	rho = 0.3101, *p* = 0.0006	rho = 0.4185, *p* = 0.0001
IL-17A (pg/mL)	rho = 0.3423, *p* = 0.0001	rho = 0.4318, *p* < 0.0001
Eotaxin (pg/mL)	rho = −0.02802, *p* = 0.76	rho = −0.0040, *p* = 0.97
G-CSF (pg/mL)	rho = 0.2586, *p* = 0.002	rho = 0.3600, *p* = 0.001
IFN-γ (pg/mL)	rho = 0.3186, *p* = 0.0004	rho = 0.2634, *p* = 0.02
IP-10 (pg/mL)	rho = 0.2289, *p* = 0.01	rho = 0.2981, *p* = 0.008
MCP-1 (pg/mL)	rho = 0.1479, *p* = 0.1084	rho = 0.1822, *p* = 0.11
MIP-1α (pg/mL)	rho = 0.2732, *p* = 0.003	rho = 0.3779, *p* = 0.0006
MIP-1β (pg/mL)	rho = 0.2953, *p* = 0.001	rho = 0.3959, *p* = 0.0003
TNF-α (pg/mL)	rho = 0.3551, *p* < 0.0001	rho = 0.4323, *p* = <0.0001

*Data is presented as rho and p values.*
*Results analysed using Pearson’s coefficient correlation test for parametric data and Spearman’s rank test for non-parametric data. ns: non-significant.*

## Data Availability

The datasets generated and/or analysed during this study are not publicly available.

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
