# Peer review of "Validation of Sputum Biomarker Immunoassays and Cytokine Expression Profiles in COPD"

_biomedicines, 2022, doi:10.3390/biomedicines10081949_

Round 1
Reviewer 1 Report
This is an interesting papers that gives useful hints to researchers that aim to use sputum analytes as biomarkers in the treatment of COPD patients, and can be the basis for future studies in the field.
I have few pint to be address
-It is not clear how the number of participant in each cohort has been chosen. Did the authors perform a power calculation to determine what would be the minimun number of participants needed?
- Abbreviation should be explained
- please verify the position of table 1: currently, the headings are separated from the columns. Please explicitate %RE (table 1 and line 233)
-please verify y axis label in figure 1 and 2, the "beta" in IL-1 beta is not readable
Author Response
-It is not clear how the number of participant in each cohort has been chosen. Did the authors perform a power calculation to determine what would be the minimun number of participants needed?
Response
We did not perform a power calculation, as the purpose of this study was to investigate variability and produce variability estimates that could be used for future power calculations. We used sample sizes that we considered sufficient to produce variability estimates, and compare between subject groups, based on previous sputum studies. We have added text to methods and discussion on this important point.
- Abbreviation should be explained
Response
We have amended the text to define all abbreviations used.
- please verify the position of table 1: currently, the headings are separated from the columns. Please explicitate %RE (table 1 and line 233)
Response
This has been corrected, and table footnotes formatted for clarity.
-please verify y axis label in figure 1 and 2, the "beta" in IL-1 beta is not readable
Response
This has been corrected.
Reviewer 2 Report
Although the manuscript is of some interest it raises some concerns
1) The design of the study is unclear and it is very difficult to identify the main question on which the study was designed. The study start out with experiments for the validation of biomarker immunoessays and then have multiple comparisons that made it difficult to determine the main message of the paper. It seems different studies were merely combined without an analysis plan. The paragraph 2.6 “study design” does not help to clarify this point.
2) In addition, the paragraph 2.8 “Sputum Supernatant biomarkers” reports that different biomarkers were examined in the two different cohorts of COPD subjects but the reason of this choice is not explain. More generally the reason for examining 2 cohorts of COPD subjects is not explain
3) Overall, the paper as presented appears as a long list of results poorly related each other, and very difficult to follow for the reader. In the introduction, it should be explained the rationale for examining in the same study subgroups of COPD patients classified on different parameters such as exacerbation, inflammatory cells, smoking status and HI infection.
4) The title refers to COPD endotypes and “endotype” is one of the key words. This is quite surprising considering that Introduction and Discussion do never refer to “endotypes”
5) Methods part 1 should report that 32 analytes were examined. This information now can be found only in the results (line227)
6) Legend of figure 4: Data are not presented as individual data
Author Response
- The design of the study is unclear and it is very difficult to identify the main question on which the study was designed. The study start out with experiments for the validation of biomarker immunoessays and then have multiple comparisons that made it difficult to determine the main message of the paper. It seems different studies were merely combined without an analysis plan. The paragraph 2.6 “study design” does not help to clarify this point.
Response
To help make this clearer, we have rewritten the end of the introduction to make the main aim of the study clearer, and explain how the two parts of the study are linked together, namely part 1 (assay validation) and part 2 (application of validated biomarkers). One of the new sentences is below:
“Our aim was to validate sputum supernatant biomarkers according to current regulatory standards and demonstrate their practical utility in COPD clinical studies”
We have also added a flow diagram to the supplement (referenced in section 2,1 – Subjects) to help visualise the different experiments and studies.
- In addition, the paragraph 2.8 “Sputum Supernatant biomarkers” reports that different biomarkers were examined in the two different cohorts of COPD subjects but the reason of this choice is not explain. More generally the reason for examining 2 cohorts of COPD subjects is not explain
Response
Sputum sampling provides limited supernatant for analysis. Therefore, we could not evaluate all the assays in one cohort. We therefore decided to use one cohort for ELISAS and the 3 plex (luminex), and a larger cohort for the 27 plex. The larger cohort also allowed bacteriology analysis. We have now explained this in the introduction, and added some discussion.
- Overall, the paper as presented appears as a long list of results poorly related each other, and very difficult to follow for the reader. In the introduction, it should be explained the rationale for examining in the same study subgroups of COPD patients classified on different parameters such as exacerbation, inflammatory cells, smoking status and HI infection.
Response
We have rewritten the introduction on these points, including explanation of the heterogeneity of COPD and how biomarkers can vary between subgroups.
- The title refers to COPD endotypes and “endotype” is one of the key words. This is quite surprising considering that Introduction and Discussion do never refer to “endotypes”
Response
Thankyou for pointing out. We have deleted endotype from the title, and this is now mentioned in the introduction
- Methods part 1 should report that 32 analytes were examined. This information now can be found only in the results (line227)
Response
We have added some text to section 2.3.
6) Legend of figure 4: Data are not presented as individual data
Response
Thank you for pointing this out, we have corrected the legends.
Reviewer 3 Report
The study assesses the suitability of several immunoassays on sputum samples of COPD patients for the validation of novel biomarkers for disease screening and staging. Sputum supernatants from patients in Cohort A (n=30 COPD, n=10 smokers, n=10 healthy) and Cohort B (n=81 COPD, n=15 smokers, n=26 healthy) were analyzed. Sample processing and analysis have been performed well. The studies are done using standard protocols. The generated data is extensive and the interpretations look accurate.
The only concern is there are several studies in the literature on cytokines done in the sputum samples of COPD and TB patients. Hence, the intent of novelty is clear. It would be helpful to clearly state the novelty in this approach over what has been published in the past.
Author Response
The only concern is there are several studies in the literature on cytokines done in the sputum samples of COPD and TB patients. Hence, the intent of novelty is clear. It would be helpful to clearly state the novelty in this approach over what has been published in the past.
Response
The main novelty is the extensive validation according to regulatory standards. We have added text to introduction to state this. The discussion contains details of our validation approach compared to previous publications. The key introduction text explaining the novelty is pasted below
“However, many of these previous studies did not perform comprehensive assay validation according to regulatory standards, including analyte recovery during sample dilution, suitability of the reference standard, precision and reproducibility. The main novelty of the work described in this manuscript is the use of regulatory standards to assess sputum supernatant assay validity in COPD. “
Round 2
Reviewer 2 Report
The authors response to my comments was satisfactory